# Piperacillin–Tazobactam Plus Vancomycin-Associated Acute Kidney Injury in Adults: Can Teicoplanin or Other Antipseudomonal Beta-Lactams Be Remedies?

**DOI:** 10.3390/healthcare10081582

**Published:** 2022-08-20

**Authors:** Abdullah Tarık Aslan, Murat Akova

**Affiliations:** 1Department of Internal Medicine, Gölhisar State Hospital, Gölhisar, 15100 Burdur, Turkey; 2Department of Infectious Diseases and Clinical Microbiology, Hacettepe University, Sihhiye, 06100 Ankara, Turkey

**Keywords:** piperacillin–tazobactam, vancomycin, teicoplanin, acute kidney injury, KDIGO

## Abstract

Numerous observational studies and meta-analyses have suggested that combination therapy consisting of piperacillin–tazobactam (TZP) and vancomycin (VAN) augments acute kidney injury (AKI) risk when compared to viable alternatives, such as cefepime–vancomycin (FEP–VAN) and meropenem–VAN. However, the exact pathophysiological mechanisms of this phenomenon are still unclear. One major limitation of the existing studies is the utilization of serum creatinine to quantify AKI since serum creatinine is not a sufficiently sensitive and specific biomarker to truly define the causal relationship between TZP–VAN exposure and nephrotoxicity. Even so, some preventive measures can be taken to reduce the risk of AKI when TZP–VAN is preferred. These measures include limiting the administration of TZP–VAN to 72 h, choosing FEP–VAN in place of TZP–VAN in appropriate cases, monitoring the VAN area under the curve level rather than the VAN trough level, avoiding exposure to other nephrotoxic agents, and minimizing the prescription of TZP–VAN for patients with a high risk of AKI. More data are needed to comment on the beneficial impact of the extended-infusion regimen of TZP on nephrotoxicity. Additionally, TZP and teicoplanin can be reasonable alternatives to TZP–VAN for the purpose of lowering AKI risk. However, the data are scarce to advocate this practice convincingly.

## 1. Introduction

Acute kidney injury (AKI) has been observed in up to a quarter of hospitalized patients and is associated with excess mortality and morbidity [1]. As a risk factor for the development of AKI in these patients, antibiotics undoubtedly play a critical role with the main offending agents, such as acyclovir, amphotericin B, aminoglycosides, colistin, and vancomycin (VAN) [2,3]. The relationship between AKI and VAN exposure has been known for a long time and was initially the result of impurities in early formulations. Owing to the technical developments in drug manufacturing, the increased nephrotoxicity risk related to early VAN formulations was eventually eliminated [4]. Nevertheless, nephrotoxicity may be augmented with several drug combinations, including piperacillin–tazobactam (TZP) plus VAN for which the incidence of AKI has been reported within a range of 5.5% to 46.0% [5]. Besides a TZP–VAN combination regimen, high VAN trough levels, concurrent exposures to other nephrotoxic medications, long duration of VAN therapy (>7 days), the severity of illness, underlying kidney dysfunction, obesity, and ICU admission are other relevant risk factors for VAN-related AKI [6]. From a pathophysiological point of view, VAN-associated AKI can be mediated by proximal tubular injury, interstitial nephritis, and cast nephropathy [7,8]. However, the mechanisms underlying the synergistic nephrotoxic interaction between TZP and VAN are still unclear.

Many retrospective cohort studies and meta-analyses have demonstrated that TZP plus VAN is associated with a higher risk of AKI than those of other VAN plus β-lactam combinations [9,10,11,12]. In a meta-analysis that included 14 observational studies, concomitant use of VAN and TZP was reported as a risk factor for increased AKI (*p* = 0.001). Intriguingly, a higher risk of AKI was detected only in those studies in which the ratio of patients receiving antibiotic therapy in ICUs was <50% (in adjusted analysis OR, 3.04; 95% CI, 1.49–6.22; *p* = 0.002) [13]. Similarly, another recent systematic review and network meta-analysis reported that the TZP–VAN combination was significantly more nephrotoxic than VAN alone or VAN in combination with meropenem (MER) or cefepime (FEP) [12].

As another parenteral glycopeptide antibiotic, teicoplanin (TEI) can well be used in place of VAN in many indications and it is widely available worldwide, including in Europe, the Middle East, and Asia-Pacific, but not in the US [14]. Previous studies comparing TEI and VAN usually indicated a safer nephrotoxicity profile with the former antibiotic [15]. In a Cochrane systematic review and meta-analysis, 24 randomized controlled trials that included 2610 patients with proven or suspected Gram-positive infections, TEI had a lower risk of nephrotoxicity than VAN (RR, 0.66; 95% CI, 0.48–0.90; I^2^ = 10%) and no patient required dialysis in either TEI or VAN group. Furthermore, clinical cure and microbiological eradication rates were similar to TEI and VAN (RR, 1.03; 95% CI, 0.98–1.08; I^2^ = 0%). However, the randomized controlled trials included in this meta-analysis were small and most of the studies had methodological problems. Therefore, the quality of the evidence regarding the risk of AKI of TEI compared to that of VAN was assessed as moderate according to the GRADE system [16].

In preparation for this article, a literature review was conducted by using PubMed/Medline, Web of Science, and Scopus databases without any date restriction. The search was undertaken until May 2022 and only articles published in English were included. It was aimed to overview contemporary data regarding the epidemiology of TZP plus VAN-associated AKI, its potential pathobiological mechanisms, and the nephrotoxicity risk of TZP–VAN as compared with that of TZP–TEI.

## 2. Epidemiology of TZP Plus VAN-Associated AKI

For the first time in the literature, the risk of AKI related with the TZP–VAN combination regimen was reported in 2011 [17]. Since then, contemporary literature has been inundated with a deluge of observational studies comparing the AKI risk of TZP–VAN with either those of VAN alone or VAN plus other antipseudomonal β-lactam agents. The TZP plus VAN combination provides a wide spectrum of activity against methicillin-resistant *Staphylococcus aureus* (MRSA), *Enterobacterales*, *Enterococcus* spp., *Pseudomonas aeruginosa*, and anaerobes; thus, the combination is typically used as empirical therapy in patients who are at risk of infections caused by these pathogens. TZP can be substituted with other antipseudomonal β-lactams, including meropenem for the same indications. In this regard, a large number of observational studies have been published comparing the rates of AKI seen in patients receiving TZP–VAN and those treated with FEP–VAN or MER–VAN. It should be noted that these studies minimize the confounding by indication that is typical when the comparator group comprises patients receiving VAN monotherapy. The results of the studies are summarized in Table 1 [9,10,18,19,20,21,22,23,24,25,26,27,28,29,30,31,32,33,34,35,36,37,38,39,40,41]. According to these studies, patients treated to the TZP–VAN combination regimen are 1.2–9.5 times more likely to develop AKI compared to those receiving FEP–VAN or MER–VAN combinations. However, these results should be cautiously evaluated due to following reasons: (I) the presence of significant heterogeneity between the comparison groups in terms of baseline characteristics of recruited patients, (II) differences in criteria used to define AKI, (III) different comparison groups (e.g., TZP–VAN vs. FEP–VAN), (IV) variations in the level of VAN exposure, (V) percentage of critically ill patients in the whole cohort, (VI) number of other nephrotoxic agents received, (VII) sample size of the studies, (VIII) statistical methodologies being used, (IX) percentage of patients with baseline kidney dysfunction within the entire cohort. In addition, although some studies performed multivariate analyses and propensity score–matched analyses, the impacts of other confounding factors not taken into account and selection bias could not be eliminated completely. Moreover, in the vast majority of the studies, since the data collections were done retrospectively and extracted in a nonblinded manner from the electronic patient records in single institutions, no causal relationships can be established. In some studies, the details of the patients’ records do not allow for evaluation of each potential risk factor for AKI, such as the Acute Physiology and Chronic Health Evaluation II (APACHE II) score, hypovolemia, hypoalbuminemia, VAN serum level, and hypotension. Moreover, the impacts of multiple generic products of antibiotics on the AKI risk should not be underestimated. Because of the retrospective nature of the studies, urine output could not be assessed for the AKI definition, which may affect the rates of AKI. Finally, in some studies, the nephrotoxic potentials of the agents were thought to be the same, but this is not true. Furthermore, the dual representation of nephrotoxic exposure does not explain the duration and dose of agents taken over the course of treatment. Therefore, this approach cannot reflect the actual exposure to other nephrotoxic agents.

Considering the absence of randomized controlled trials comparing the risk of AKI with TZP–VAN and FEP–VAN or MER–VAN, meta-analyses evaluating the same pool of observational studies may only serve to amplify bias. Nevertheless, seven meta-analyses have been reported to address the relationship between TZP–VAN and AKI as depicted in Table 2 [5,12,13,42,43,44,45]. Hammond et al. conducted a meta-analysis that included 14 observational studies and showed that TZP–VAN was significantly associated with a higher rate of AKI compared to FEP–VAN or MER–VAN in adults (the adjusted odds ratio (OR, 3.15; 95% CI, 1.72–5.76) [13]. However, it is noteworthy that substantial statistical heterogeneity was found among the studies (I^2^ = 78.1%). In another meta-analysis, Giuliano et al. evaluated 15 observational studies, 7 of which overlapped with the studies included in the meta-analysis by Hammond et al. [5]. The authors demonstrated considerable risk for AKI with TZP–VAN compared to vancomycin with or without another β-lactam (OR, 3.649; 95% CI, 2.157–6.174; I^2^ = 83.5%; *p* < 0.001) [5]. Furthermore, this association remained significant when the TZP–VAN combination was compared to VAN alone (OR, 3.980; 95% CI, 2.749–5.763; I^2^ = 31.4%; *p* < 0.001). In a recent meta-analysis (47 cohort studies with a total of 56,984 adult and pediatric patients), TZP–VAN was significantly associated with higher odds of AKI than vancomycin monotherapy (OR, 2.05; 95% CI, 1.17–3.46) and its concomitant use with meropenem (OR, 1.84; 95% CI, 1.02–3.10) or cefepime (OR, 1.80; 95% CI, 1.13–2.77) [12]. On the other hand, all secondary outcomes, including the severity of AKI, time to AKI, duration of kidney injury, the need for renal replacement therapy, length of hospitalization, and mortality were similar between the comparison groups. In this meta-analysis, the level of evidence was interpreted to be moderate, mainly because of the presence of inter-study heterogeneity as a consequence of the methodological differences of the included studies. The power of the outcomes was strengthened by performing a sensitivity analysis, which indicated that TZP–VAN was the most nephrotoxic combination regimen when only studies at low risk of bias were analyzed.

The definition used to define AKI varies significantly between the studies. Some studies use acute kidney injury network (AKIN) and kidney disease improving global outcomes (KDIGO) criteria, which include more AKI cases with smaller serum creatinine elevations (>0.3 mg/dl) than the RIFLE (risk, injury, failure, loss of kidney function, and end-stage kidney disease) criteria, which require at least ≥50% increment in the serum creatinine level to quantify the presence of AKI. Therefore, selected AKI definition criteria seem to impact the incidence of stage 1 AKI without affecting the frequency of stage II or III AKI [46]. It is important to underline that TZP–VAN-associated AKI is generally mild in severity (stage I AKI or risk class of the RIFLE criteria). The incidence of severe AKI requiring renal replacement therapy is not significantly higher in the TZP–VAN group compared to other groups [27,39]. Similarly, studies that included ICU patients indicated that there was no incremental risk of either persistent kidney dysfunction or requirement of renal replacement therapy for patients receiving TZP–VAN over those receiving FEP–VAN or MER–VAN [12,21]. Although the TZP–VAN combination does not seem to increase the risk of severe AKI (stage II or stage III AKI or requirement of RRT) over other comparators, even stage I AKI can dramatically reduce long-term survival rates, increase morbidity, prolong hospitalizations, and ramp up healthcare-related costs [47,48]. Taken together, the TZP–VAN combination appears to be frequently associated with mild (stage 1) AKI in critically and non-critically ill patients. The clinical importance of stage I AKI should not be underestimated as it is significantly associated with adverse clinical and economic consequences.

## 3. Epidemiology of TZP Plus VAN-Associated AKI in ICU Patients

Although many observational studies have included ICU patients as part of the entire cohort, eight studies have investigated the risk of AKI only in ICU patients receiving TZP–VAN compared to patients receiving FEP–VAN or those receiving FEP–VAN or MER–VAN. All these studies have retrospective single-center designs with sample sizes ranging from 122 to 3299. Except for two studies (one from South Korea and the other from Australia), all were published in the USA. Among them, Blevins et al. reported that the AKI rates were 39.3% for TZP–VAN patients, 24.2% for FEP–VAN patients, and 23.5% for MER–VAN patients (*p* < 0.0001 for both comparisons). Similarly, the frequencies of stage II and stage III AKI were also significantly higher for TZP–VAN patients than for other patients receiving MER–VAN or FEP–VAN (15% and 6.6% for TZP–VAN patients, 5.8% and 1.8% for FEP–VAN patients, and 6.6% and 1.3% for MER–VAN patients, *p* < 0.0001 for both comparisons). In a multivariate analysis, utilization of TZP–VAN was found to be an independent risk factor of AKI (OR, 2.161; 95% CI, 1.62–2.88) [33]. In line with these results, Kang et al. revealed an increased risk of AKI in the TZP–VAN group in comparison with the FEP–VAN group (52.7% vs. 27.7%, *p* < 0.001) in 340 ICU patients [34]. In other studies (n = 6), although the incidences of AKI were higher in the TZP–VAN patients than in the comparison groups numerically, these differences were not able to attain statistical significance [20,30,32,35,37,41]. Similarly, in a meta-analysis, Hammond et al. showed that a higher risk of AKI was not observed in the TZP–VAN group when the studies with ≥50% of patients receiving antibiotic therapy in ICUs were included in the analysis alone (in adjusted analysis OR, 2.83; 95% CI, 0.74–10.85) [13]. In another meta-analysis, Luther et al. conducted a subanalysis of critically ill patients (n = 968) and the odds of AKI in the TZP–VAN group were not significantly different from those of the FEP–VAN or MER–VAN groups (odds ratio, 1.43; 95% CI, 0.83–2.47) [43]. Consistently, Bellos et al. indicated that concomitant administration of TZP and VAN had the highest probability of AKI as compared to other groups in a separate analysis of ICU patients (i.e., VAN monotherapy, FEP–VAN, and MER–VAN). However, the results did not reach statistical significance when compared with other combinations [12]. It is unclear why a statistically significant difference in AKI risk could not be obtained in ICU patients in those receiving TZP–VAN compared to other comparison groups. Nevertheless, some specific risk factors prevailingly seen in ICU patients, such as critical illness, hypotension, and exposure to vasopressors, may have precluded us to uncover the real impact of TZP–VAN exposure on the risk of AKI. More data are needed to clarify the precise pathophysiological mechanism(s) for the reasons of the non-significant association between TZP–VAN exposure and AKI compared to VAN plus FEP or MER in ICU patients.

## 4. How Can We Reduce the Risk of AKI due to TZP–VAN Exposure?

In many countries, vancomycin is overly prescribed for inappropriate indications, especially community-acquired infections [49]. Similarly, antipseudomonal β-lactam agents, particularly TZP, are given frequently in nosocomial infections with inadequate assessments of risk factors for multidrug-resistant Gram-negative microorganisms [50]. The risks and benefits must be interpreted carefully when choosing empirical antimicrobial therapy and minimizing unnecessary utilization of broad-spectrum antibiotics. A wide range of strategies has been suggested to alleviate AKI risk following TZP–VAN administration. In addition to classical measures, including close monitoring of renal function, adequate hydration, avoidance of other nephrotoxic agents, and limiting the administration of empirical VAN therapy by excluding MRSA nasal colonization via polymerase chain reaction testing, some additional measures can be advocated to reduce the risk of AKI from TZP–VAN exposure.

### 4.1. Restricting the Use of TZP–VAN Combination Therapy for More Than 72 h

Rapid onset of AKI in response to TZP–VAN vs. other comparators can be considered as a parameter-strengthening specific association of TZP–VAN with nephrotoxicity. In a retrospective single-center study, Navalkele et al. showed that the onset of AKI was more rapid in patients receiving TZP–VAN (median duration 3 days, interquartile range [IQR], 2–5 days) as compared to those treated with FEP–VAN (median duration 5 days, IQR, 3–7 days) [23]. Another study demonstrated the highest daily incidence of AKI occurred on day 5 of TZP–VAN therapy [46]. Therefore, it seems reasonable that one way to mitigate the risk of nephrotoxicity caused by TZP–VAN combination therapy is early and regular assessment of the regimen with a goal of rapid de-escalation. However, studies demonstrating nephroprotective effects of early antibiotic discontinuation were scarce, since those receiving the investigated treatment regimens concurrently for <48 h were typically excluded from the vast majority of studies. Nevertheless, Lorentz et al. suggested that the implementation of a 72-h restriction program for administration of TZP significantly shortened the exposure time to this antibiotic, thus resulting in reduced rates of TZP–VAN-associated AKI [51]. In line with this study, Schreier et al. compared the risk of AKI with a brief course of TZP–VAN (24–72 h) with the risk associated with other antipseudomonal β-lactam plus VAN combinations in a single-center retrospective cohort study that included 3299 ICU patients [32]. As a result, the authors demonstrated that a short course of TZP–VAN therapy did not confer a higher risk of stage II or III AKI after adjustment for relevant confounders (adjusted odds ratio [95% confidence interval] TZP–VAN vs. FEP–VAN, 1.11 [0.85–1.45]; TZP–VAN vs. MER–VAN, 1.04 [0.71–1.42]). Similarly, a retrospective single-center cohort study indicated that a short-course TZP–VAN regimen (24–60 h) was significantly associated with a lower risk of AKI as compared with extended-course TZP–VAN (>72 h) therapy [52]. Consequently, antimicrobial stewardship practices minimizing administration of TZP–VAN beyond 72 h seem to be plausible.

### 4.2. Administration of TZP as an Extended Infusion Regimen

In the literature, some recent investigations have pointed out the potential benefit of extended infusion of TZP to obtain more favorable clinical outcomes, especially in the setting of sepsis and febrile neutropenia [53,54,55]. In contrast, the findings of the meta-analysis by Bellos et al. indicated an insignificant propensity of vancomycin combined with extended-infusion TZP towards lower nephrotoxicity risk (SUCRA: 51.5% vs. 79%); statistical significance was not attained and, thus, the renoprotective effect of this strategy was not assured [12]. Similarly, several retrospective cohort studies did not show any difference in AKI rates between patients receiving extended-infusion TZP plus VAN and patients treated with standard infusion TZP plus VAN [47,56,57]. Nevertheless, given the superior clinical outcomes related to the TZP extended-infusion regimen and the findings of the meta-analysis by Bellos et al., the administration of extended-infusion TZP might be suggested to reduce the incidence of AKI until randomized controlled trials show otherwise.

### 4.3. Application of Area under Curve (AUC)-Guided VAN Dosing

Several retrospective studies have attempted to elucidate the link between VAN exposure and the probability of AKI [58,59]. The existing studies suggest that the trough concentration of VAN above 15 to 20 mg/L significantly increases the risk of AKI [60]. In parallel with this finding, some recent studies indicated that the risk of AKI augments along the vancomycin AUC continuum, particularly when the daily AUC is more than 650 mg·h/L [58,59,61]. Moreover, animal studies went a step further and showed that elevated vancomycin AUC rather than the trough concentration is a more reliable predictor of AKI [62,63]. In a retrospective cohort study, patients with AUC values between 600 and 800 mg·h/L were more likely to develop AKI as compared to those having AUC levels between 400 and 600 mg·h/L (*p* = 0.014) [58]. Similarly, Lodise et al. showed that the probability of AKI increased 2.5-fold in patients with daily AUC levels above 1300 mg·h/L compared to those with lower values (30.8% vs. 13.1%, *p* = 0.02) [59]. Furthermore, the researchers investigated which AUC-guided and trough-guided vancomycin dosing might better predict the risk of AKI. In a large-scale retrospective, quasi-experimental study, Finch et al. looked at the incidence of AKI in patients monitored by individualized AUC vs. trough concentrations. In this study, AUC-guided VAN dosing was reported as an independent protective factor for AKI (odds ratio [OR], 0.52; 95% CI, 0.34–0.80; *p* = 0.003) [64]. According to contemporary literature, daily vancomycin AUC values should be kept between 400 and 600 mg·h/L to minimize the risk of nephrotoxicity. On the other hand, it is not well-known what the safe vancomycin AUC threshold should be in case of concomitant exposure to TZP. A single-center, retrospective, pre–post-quasi-experimental study showed no significant difference in the incidence of AKI between patients receiving TZP–VAN based on trough-guided VAN dosing and those receiving TZP–VAN with AUC-guided VAN dosing (17.8% vs. 13.6%; *p* = 0.371) [65]. In conclusion, future studies should investigate what the optimal threshold for the AUC-guided VAN dosing strategy should be in patients receiving TZP–VAN combination therapy.

## 5. Pathophysiological Mechanisms of TZP Plus VAN-Associated AKI

Even though many observational studies and seven meta-analyses supported the association between TZP–VAN and the increased risk of AKI, to date, none of these studies has proved biological plausibility. The first issue is that all studies have relied on serum creatinine rise to define AKI. However, it is well known that changes in serum creatinine levels are not sufficiently sensitive and specific for defining AKI and may also lead to misdiagnoses [66]. For instance, glomerular filtration rates can diminish up to 50% before elevations in creatinine are observed [67]. As the serum creatinine level is determined by tubular secretion and the reabsorption capacity of the kidney, competitive drug–drug interactions can interfere with tubular secretion of creatinine and may result in an increase of the serum creatinine level despite the absence of an actual renal injury. As seen in patients receiving trimethoprim–sulfamethoxazole, an increase in serum creatinine does not always indicate the presence of real kidney damage [68,69]. Since creatinine is a typical surrogate marker of glomerular filtration, alterations in the secretory and reabsorption functions of glomeruli can change the serum concentration of creatinine while renal function remains stable. TZP–VAN-mediated increase in serum creatinine could potentially be related to a mechanism involving specific anion transporters in the renal tubules. TZP is a potent substrate for specific organic anion transporters (e.g., OAT1 and OAT3) [70,71]. The same transporters also mediate creatinine transit [72]. On the other hand, VAN inhibits messenger RNA expressions of OAT1 and OAT3 [73]. The synergistic interaction between TZP and VAN for serum creatinine rise may stem from a decrease in the number of available pumps and competition of creatinine and TZP at the organic anion transporter level (Figure 1). For correct quantification of kidney injuries in patients with TZP–VAN exposure, monitoring the urine output and evaluating other indicators of kidney functioning can be recommended.

Given the limitations of serum creatinine, several biomarkers are recommended over serum creatinine by regulatory agencies to better understand both renal injury and function. Of these biomarkers, kidney injury molecule 1 (KIM-1) and osteopontin were supported by both the US Food and Drug Administration (FDA) and the European Medicines Agency (EMA). In addition, urinary KIM-1 and clusterin were demonstrated as reliable biomarkers of drug-induced AKI [74]. These biomarkers are highly sensitive and can predict histopathological changes in renal parenchyma very rapidly [75]. Vaidya et al. showed that KIM-1 could begin to increase hours after mild tubular injury, reach a maximum at 24 h, and remain elevated for 120 h from the time of AKI [76]. In a recent study that included critically ill patients, renal stress was evaluated using urinary biomarkers [77]. The kinetics of urinary metalloproteinase 2 (TIMP-2) and insulin growth factor–binding protein 7 (IGFPB7) were compared in vancomycin alone, TZP alone, and TZP–VAN groups. It was revealed that patients receiving TZP–VAN released AKI biomarkers significantly higher than those treated by TZP or VAN monotherapy. Nevertheless, the results of this study should be evaluated cautiously since the patients receiving TZP–VAN were more critically ill as compared with both TZP and VAN monotherapy groups.

Based on another hypothesis, AKI may be derived from subclinical interstitial nephritis caused by TZP that is magnified by the oxidative stress (reactive oxygen species production) induced by VAN [78,79]. In fact, it is well known that VAN induces mitochondrial dysfunction, proinflammatory oxidative stress, and tubular cell apoptosis [6]. Consistently, VAN-related tubular injury was confirmed by elevations in NGAL and other biomarkers in several animal and human models [7,80]. In contrast, animal or human models demonstrating TZP–VAN-associated AKI using urinary biomarkers or histopathological examination are scarce in the literature. Nevertheless, in a case report form, a 16-year-old patient with acute leukemia was treated with TZP–VAN, and a biopsy specimen taken 23 days after the onset of hemodialysis support showed the presence of tubulointerstitial nephritis, tubular damage, and interstitial edema [81]. There are two more case reports demonstrating acute interstitial nephritis and acute tubular necrosis in the TZP–VAN-receiving patients [82,83]. On the other hand, it should be underscored that interstitial nephritis is an extremely rare event in our daily practice. Therefore, given the high frequency of AKI seen after TZP–VAN exposure, it is unlikely that a significant increase in serum creatinine occurs after exposure to TZP–VAN combination therapy through an interstitial nephritis-associated mechanism [43]. Another theory is that TZP might hamper the effective clearance of VAN and, thereby, lead to VAN accumulation in the nephron [84]. However, there is no adequate evidence to support this hypothesis.

In animal models investigating TZP–VAN-related AKI, a combination regimen was not found worse than VAN alone [85]. Pais et al. showed that histopathological kidney injury score is similar in the TZP alone group and TZP–VAN group (*p* = 0.76) but rats receiving VAN monotherapy had elevated scores as compared with the TZP alone group (*p* = 0.044). These authors also demonstrated that the level of urinary biomarkers started to increase 24 h after VAN exposure but it did not increase in the TZP–VAN group until day 3 [85]. In the same study, in NRK-52E cells, VAN resulted in cell death with high doses (IC50 48.76 mg/mL) but TZP did not, and TZP–VAN was similar to VAN alone.

In summary, we need more studies to determine the causality of TZP–VAN exposure for the increased risk of AKI. The pathophysiological mechanism of TZP–VAN-associated AKI, if any, remains elusive. Furthermore, relying on serum creatinine as a surrogate of glomerular function may be associated with misleading results.

## 6. Comparison of TZP–TEI and TZP–VAN Regimens in Terms of AKI Risk

Given the increasing risk of AKI with TZP–VAN combination therapy, some authors explored the regimens containing an antipseudomonal β-lactam plus VAN combinations (e.g., FEP–VAN and MER–VAN) as alternatives to TZP–VAN. In these studies, TZP–VAN was found to be consistently associated with a higher incidence of AKI as compared with FEP–VAN and MER–VAN in non-critically ill patients. In critically ill patients, the results are conflicting. As another alternative to TZP–VAN combination therapy, the authors from some European and Asian countries looked at the risk of AKI with TZP–TEI combination regimen compared to either TEI monotherapy or TZP–VAN combination therapy. Considering the potent anti-MRSA activity of TEI, TZP–TEI can be a reasonable alternative to TZP–VAN if the incidence of AKI is lower with this regimen. With respect to this point of view, we compared the rates of AKI, 7-day and 30-day mortalities, and resolution of AKI at discharge in patients receiving TZP–TEI vs. TZP–VAN in a single-center, retrospective cohort study [39]. The AKI was defined per RIFLE criteria. In a multivariate analysis of the entire cohort, TZP–VAN was found to be associated with a significantly higher rate of AKI as compared with TZP–TEI (aOR: 3.21, 95% CI, 1.36–7.57; *p* = 0.008) or with MER–VAN (aOR: 2.28, 95% CI, 1.008–5.18; *p* = 0.048). In a multivariate analysis of the matched cohort, TZP–VAN had 3.96 odds of AKI (95% CI, 1.48–10.63, *p* = 0.006) as compared with TZP–TEI. Seven-day and thirty-day mortalities and resolution rates of AKI were similar in both groups [39]. In contrast, Shao et al. compared TZP–VAN (n = 211) and TZP–TEI (n = 211) in terms of AKI risk in a 1:1 propensity score–matched analysis. In this study, the risk of AKI in the TZP–TEI group was similar to that in the TZP–VAN group (12.3% vs. 11.4%; HR, 1.25 [0.72–2.18]; *p*: 0.44) [86]. Nevertheless, TZP–TEI may increase the risk of AKI compared to TEI alone or TZP alone. In a study from the Netherlands, among 4202 patients, 3188 (75.9%) were treated with TZP alone, 791 (18.8%) received TEI monotherapy, and 223 (5.3%) received a TZP–TEI combination regimen [87]. The incidence of AKI was 5.4% for TZP alone, 3.4% for TEI, and 11.7% for TZP–TEI (*p* < 0.001). After correcting for confounding factors via a multiple logistic regression analysis, the same pattern remained unchanged. Additionally, mean serum creatinine tested at 48–72 h of initiation of therapy was slightly higher in the TZP–TEI group as compared with the baseline [+1.61% (95% CI −2.25 to 5.70)], indicating only a slight increase in the serum creatinine level [87]. According to these results, although TZP–TEI is associated with a higher prevalence of AKI compared with either TZP or TEI monotherapy, the overall increment in the serum creatinine level with TZP–TEI is very small. In another study from Taiwan, Tai et al. conducted a 1:3 propensity score–matched analysis involving 954 patients (243 pairs in total) receiving either TZP–TEI or TEI plus another antipseudomonal β-lactam [88]. The patients receiving TZP–TEI did not differ significantly from those treated with TEI plus another antipseudomonal β-lactam in terms of AKI risk (14.8% versus 14.2%; *p* = 0.815). However, the time to AKI was significantly shorter in the TZP–TEI group (4.64 ± 2.33 versus 6.29 ± 4.72 days; *p* = 0.039) [88].

Since the existing literature is highly heterogenous and only a few retrospective single-center cohort studies have compared TZP–VAN and TZP–TEI in the context of AKI, we started to enroll patients in a prospective multicenter multinational cohort study entitled ‘Comparison of piperacillin–tazobactam and vancomycin with piperacillin–tazobactam and teicoplanin for the risk of acute kidney injury (CONCOMITANT)’. In this study, we not only compare the risk of AKI in both groups but also analyze the impact of TZP dose, VAN dose, VAN AUC level, TEI serum trough level, and extended infusion TZP regimen on AKI risk by a multivariate analysis and propensity score–matched analysis. The primary outcome is to compare the rate of AKI occurring between the first day of antibiotic treatment and the third day after completing therapy according to KDIGO criteria. The secondary outcomes are as follows: 7-day mortality, 30-day mortality, renal function status at discharge (resolved, the injury still present, the requirement of renal replacement therapy), and time to AKI. Patients are being included in the study if they are ≥18 years of age, have a baseline serum creatinine level measured within one week of antibiotic initiation and receive antibiotic combinations in the investigation for at least 48 h concomitantly, and have the two antibiotics initiated within 48 h of each another. For patients who receive multiple courses of antibiotic therapy during hospitalization, only the initial course is included. Patients are excluded if they are receiving renal replacement therapy (RRT), are pregnant or breastfeeding, and are in palliative care at the time of antibiotic initiation. Other reasons for exclusion are receiving antibiotic therapies for less than 48 h, receiving antibiotics at separate times, having a baseline GFR < 60 mL/min or serum creatinine level >1.2 mg/dl, developing AKI, or undergoing dialysis prior to commencement of TZP–TEI or TZP–VAN, having incomplete data from either their hospital admission or from the follow-up period, and having a diagnosis of cystic fibrosis. In this study, the patient recruitment process will be continued up until 1 August 2023.

## 7. Conclusions

A large number of observational studies and meta-analyses have supported the link between nephrotoxicity and the concurrent use of TZP and VAN as compared with VAN alone and VAN plus other antipseudomonal β-lactam antibiotics. However, the pathophysiological mechanisms of AKI caused by the TZP–VAN combination therapy have not been clearly revealed yet. In addition, using a potentially flawed marker of AKI (e.g., serum creatinine) may have led to incorrect findings. It is clear that human and animal data are needed to confirm the causality between TZP–VAN and AKI using more reliable kidney injury biomarkers and histopathological examinations. Although it is uncertain whether TZP–VAN truly increases the risk of AKI, some specific measures can be taken to reduce the nephrotoxicity risk. Additionally, we need more reliable data to recommend TZP–TEI over TZP–VAN for the purpose of declining the frequency of AKI.

## Figures and Tables

**Figure 1 healthcare-10-01582-f001:**
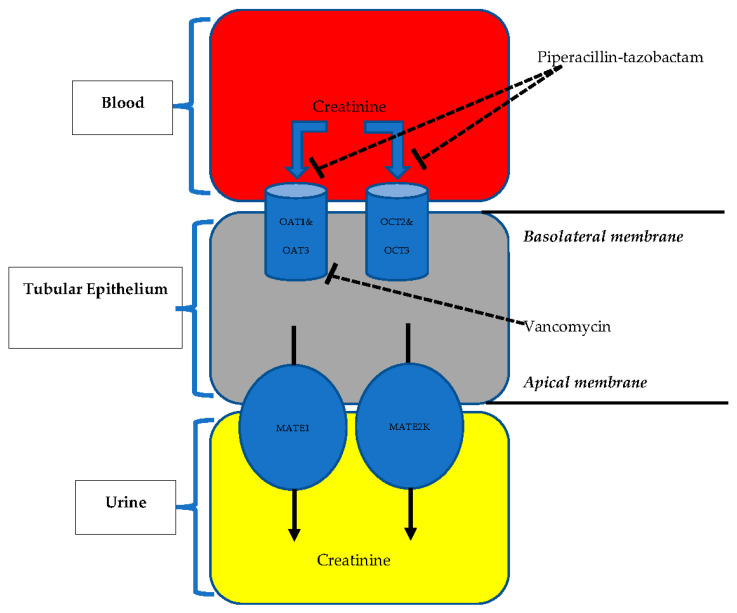
Possible mechanism of the increase in serum creatinine associated with piperacillin–tazobactam and vancomycin combination. Secretion of creatinine into the tubular lumen is partially mediated by OAT1–OAT3 and OCT2–OCT3 transporters and MATE1–MATE2K. Piperacillin–tazobactam has a significant binding affinity for OAT1 and OAT3 and limits the transport of creatinine into the tubular cell. Vancomycin inhibits the expression of mRNA involved in OAT1–OAT3 externalization. Abbreviations: MATE, multidrug and toxin extrusion; OAT, organic anion transporter; OCT, organic cation transporter.

**Table 1 healthcare-10-01582-t001:** Studies comparing the rate of AKI with piperacillin–tazobactam plus vancomycin and meropenem or cefepime plus vancomycin.

Authors and Type	Year	Country	Population	Definition of AKI *	ICU Residence and/or Critically Ill, %	Sample Size, *n*	Exposure to Other Nephrotoxins, %	Mean or Initial VAN Trough Level (mg/dl)	Treatment Duration, Days	Comparison Groups	Rate of AKI
Moenster RP, et al. R, SC, [18]	2014	USA	Adult patients with or without renal dysfunction	RIFLE	Not provided	139	Yes, percentage unknown	15.8 vs. 14.5	14.7 vs. 11.3	TZP–VAN vs. FEP–VAN	29.3% vs. 13.3%; OR, 3.45 (0.96–12.4); *p*: 0.05
Gomes DM, et al. R, SC, [19]	2014	USA	Adult patients without renal dysfunction	AKIN	34.8 vs. 53.6	224	Yes, percentage unknown	14.1 vs. 13.06	7.1 vs. 6.7	TZP–VAN vs. FEP–VAN	34.8% vs. 12.5%; OR, 3.74 (1.89–7.39); *p*: <0.001
Hammond DA, et al. R, SC, [20]	2016	USA	Adult patients without renal dysfunction	AKIN	100	122	Yes, percentage unknown	17.9 vs. 15.1	Not provided	TZP–VAN vs. FEP–VAN	32.7% vs. 28.8%; *p*: 0.76
Al Yami MS, et al. R, MC, [21]	2017	Saudi Arabia and USA	Adult patients without renal dysfunction	KDIGO	17.6 vs. 17.3	183	62.9 vs. 46.6	15.7 vs. 16.9	4.3 vs. 5.4	TZP–VAN vs. MER–VAN	7.4% vs. 5.3%; *p*: 0.4
Rutter WC, et al. R, SC, [9]	2017	USA	Adult patients with or without renal dysfunction	RIFLE	Not provided	4193	60.7 vs. 59.4	Percentage of >20 mg/L 30.4% vs. 27.4%	3.0 vs. 4.0	TZP–VAN vs. FEP–VAN	21.4% vs. 12.5%; OR, 2.18 (1.64–2.94); *p*: < 0.001
Jeon N, et al. R, SC, [22]	2017	USA	Adult patients with or without renal dysfunction	KDIGO	14.09 vs. 18.75	5335	Yes, percentage unknown	Percentage of >20 mg/L 2.5% vs. 1.9%	5.0 vs. 5.0	TZP–VAN vs. FEP–VAN	19.6% vs. 16.2%; aHR, 1.25 (1.11–1.42); *p*: < 0.05
Navalkele B, et al. R, SC, [23]	2017	USA	Adult patients without renal dysfunction	RIFLE and AKIN	21 vs. 23	558	Yes, percentage unknown	17.3 vs. 17.7	Not provided	TZP–VAN vs. FEP–VAN	29% vs. 11%; HR, 4.27 (2.73–6.68); *p*: <0.001
Peyko V, et al. P, SC, [24]	2017	USA	Adult patients with or without renal dysfunction	KDIGO	Not provided	85	33.9 vs. 38.5	16.6 vs. 18.3	Not provided	TZP–VAN vs. MER–VAN or FEP–VAN	37.3% vs. 7.7%; *p*: 0.005
Cannon JM, et al. R, SC, [25]	2017	USA	Adult patients without renal dysfunction	RIFLE	15.8 vs. 31.1	366	Yes, percentage unknown	Percentage of >20 mg/L 21.9% vs. 28.4%	Not provided	TZP–VAN vs. MER–VAN	25.3% vs. 9.5%; *p*: 0.008
Clemmons AB, et al. R, SC, [26]	2018	Georgia	Adult patients with or without renal dysfunction	KDIGO	Not provided	170	Not provided	Percentage of >20 mg/L 42.9% vs. 31.6%	4.0 vs. 4.0	TZP–VAN vs. FEP–VAN	68% vs. 27%; OR, 5.1 (2.5–10.5); *p*: < 0.001
Mullins BP, et al. P, MC, [27]	2018	USA	Adult patients without renal dysfunction	RIFLE	34 vs. 41	242	Yes, percentage unknown	16.3 vs. 15.2	5.4 vs. 6.4	TZP–VAN vs. MER–VAN or FEP–VAN	29.8% vs. 8.8%; OR, 6.6 (2.8–15.8), *p*: <0.001
Robertson AD, et al. R, SC, [28]	2018	USA	Adult patients without renal dysfunction	RIFLE	0	169	81.2 vs. 83.3	Percentage of >20 mg/L 21.2% vs. 19.0%	4.6 vs. 4.7	TZP–VAN vs. MER–VAN	16.5% vs. 3.6%; OR, 6.8 (1.5–0.9); *p*: 0.009
Balcı C, et al. R, SC, [29]	2018	Turkey	Adult patients with or without renal dysfunction	AKIN	Not provided	132	52.8 vs. 65.2	Not provided	Not provided	TZP–VAN vs. MER–VAN	41.3% vs. 10.1%; OR, 0.33 (0.21–0.77); *p*: <0.001
Buckley MS, et al. R, SC, [30]	2018	USA	Adult patients with or without renal dysfunction	RIFLE	100	333	Yes, percentage unknown	13.5 vs. 13.1	5.1 vs. 5.8	TZP–VAN vs. FEP–VAN	19.5% vs. 17.3%; OR, 0.86 (0.49–1.53); *p*: 0.6
Rutter WC, et al. R, SC, [10]	2018	USA	Adult patients with or without renal dysfunction	RIFLE	Not provided	10,236	Yes, percentage unknown	Not provided	5.0 vs. 5.0	TZP–VAN vs. MER–VAN	27.4% vs. 15.4 %; OR, 2.53 (1.82–3.52); *p*: < 0.001
Ide N, et al. R, SC, [31]	2019	Japan	Adult patients with or without renal dysfunction	KDIGO	0	82	Yes, percentage unknown	Percentage of >15 mg/L 52.0% vs. 50.0%	Not provided	TZP–VAN vs. MER–VAN	33.3% vs. 9.1%; *p*: 0.015
Schreier DJ, et al. R, SC, [32]	2019	USA	Adult patients with or without renal dysfunction	AKIN	100	3299	Yes, percentage unknown	Not provided	All patients received 24-72 h combination therapy	TZP–VAN vs. MER–VAN vs. FEP–VAN	1.04 (0.71–1.42); *p*: 0.84 1.11 (0.85–1.45); *p*: 0.44
Blevins AM, et al. R, SC, [33]	2019	USA	Adult patients with or without renal dysfunction	KDIGO	100	2492	76.0 vs. 82.7 vs. 78.0	12.0 vs. 12.0 vs. 11.6	4.0 vs. 3.0 vs. 3.0	TZP–VAN vs. MER–VAN vs. FEP–VAN	39.3% vs. 23.5% vs. 24.2%; OR, 2.16 (1.62–2.88); *p*: < 0.001
Kang S, et al. R, SC, [34]	2019	South Korea	Adult patients with or without renal dysfunction	KDIGO	100	340	Yes, percentage unknown	Not provided	6.5 vs. 8.0 vs. 8.0	TZP–VAN vs. MER–VAN vs. VAN	52.7% vs. 27.7% vs. 25.7%; *p*: <0.001
Molina KC, et al. R, SC, [35]	2019	USA	Adult patients without renal dysfunction	AKIN	100	394	Yes, percentage unknown	11.2 vs. 11.0	3.3 vs. 3.7	TZP–VAN vs. FEP–VAN	28.7% vs. 21.3%; OR, 1.50 (0.88–2.57); *p*: 0.13
Haruki Y, et al. R, SC, [36]	2020	Japan	Adult patients without renal dysfunction	RIFLE	25.0 vs. 28.3	272	68.5 vs. 67.8	13.3 vs. 13.4	6.0 vs. 7.0	TZP–VAN vs. VAN-Other β-lactams	25.0% vs. 12.2%; OR, 2.40 (1.20–4.78); *p*: 0.01
O’ Callaghan K et al. R, SC, [37]	2020	Australia	Adult patients with or without renal dysfunction	AKIN	100	260	Yes, percentage unknown	Not provided	4.0 vs. 5.0	TZP–VAN vs. MER–VAN or FEP–VAN	RRR, 2.2 (1.0–4.9); *p*: 0.05
Yabes JM, et al. R, SC, [38]	2021	USA	Adult patients without renal dysfunction	RIFLE and AKIN	88.5 vs. 93.7	268	Yes, percentage unknown	9.4 vs. 10.9	Not provided	TZP–VAN vs. VAN-Other β-lactams	13.1% vs. 9.7%; OR, 1.72 (1.02–2.76); *p*: 0.04
Aslan AT, et al. R, SC, [39]	2021	Turkey	Adult patients with or without renal dysfunction	RIFLE	32.0 vs. 34.6	154	Yes, percentage unknown	Not provided	5.0 vs. 9.0	TZP–VAN vs. MER–VAN	40.0% vs. 24.0%; aOR, 2.28 (1.01–5.18); *p*: 0.048
Tookhi RF, et al. R, SC, [40]	2021	Saudi Arabia	Adult patients without renal dysfunction	KDIGO	18.2 vs. 30.9	158	49.4 vs. 51.9	Not provided	Not provided	TZP–VAN vs. MER–VAN	10.4% vs. 21.0%; *p*: 0.07
Elliott BP, et al. R, SC, [41]	2022	USA	Adult patients with sepsis	KDIGO	100	418	Yes, percentage unknown	Not provided	Not provided	TZP–VAN vs. FEP–VAN	15.2% vs. 11.0%; *p*: 0.44

Abbreviations: R, retrospective; SC, single-center; P, prospective; MC, multi-center; AKI, acute kidney injury; n, number; ICU, intensive care unit; VAN, vancomycin; TZP, piperacillin–tazobactam; FEP, cefepime; MER, meropenem; OR, odds ratio; aOR, adjusted odds ratio; HR, hazard ration; aHR, adjusted hazard ratio; RRR, relative risk reduction. * For definitions of AKI, please see text.

**Table 2 healthcare-10-01582-t002:** Meta-analyses evaluating the relationship between piperacillin–tazobactam plus vancomycin and higher nephrotoxicity risk.

Authors	Year	Total Number of Studies, n	Total Number of Patients	Deadline for Inclusion of Studies	Comparison Groups	The Risk of AKI	Additional or Secondary Results
Giuliano et al., [5]	2016	15 (only studies including adult patients)	3258	1 June 2016	TZP–VAN vs. VAN alone TZP–VAN vs. VAN+ β-lactam TZP–VAN vs. VAN alone or VAN+ another antibiotic	TZP–VAN vs. VAN ± β-lactam: OR, 3.65; 95% CI, 2.15–6.17; I^2^ = 83.5%, *p* < 0.001	Abstracts were removed: OR, 3.498; 95% CI 1.747–7.003, I^2^ = 82.3%, *p* < 0.001) Low-quality studies were removed: OR, 4.596; 95% CI 2.929–7.212; I^2^ = 0%, *p* < 0.001).
Hammond DM, et al., [13]	2017	14 (11 included only adults and 3 included only children)	3549	October 2016	TZP–VAN vs. VAN alone TZP–VAN vs. FEP–VAN TZP–VAN vs. VAN+ β-lactam	In adults: aOR, 3.15; 95% CI, 1.72–5.76 In children: OR, 4.55; 95% CI, 2.71–10.21	<50% of patients received care in an ICU: aOR, 3.04; 95% CI, 1.49–6.22 ≥50% of patients received care in an ICU: aOR, 2.83; 95% CI, 0.74–10.85
Chen et al., [42]	2018	8 (7 included only adults and 1 included only children)	10,727	April 2017	TZP–VAN vs. VAN+ β-lactam TZP–VAN vs. FEP–VAN TZP–VAN vs. VAN	TZP–VAN vs. VAN+ β-lactam: OR, 1.57; 95% CI, 1.13–2.01; I^2^ = 76.4%, *p* < 0.001	TZP–VAN vs. FEP–VAN: OR, 1.50; 95% CI, 1.07–1.93; I^2^ = 80.5%, *p* < 0.001 TZP–VAN vs. VAN: OR, 1.49; 95% CI, 1.06–1.92; I^2^ = 84.1%, *p* < 0.001
Luther et al., [43]	2018	32 (Only studies including adult patients)	24,799	April 2017	TZP–VAN vs. VAN alone TZP–VAN vs. FEP–VAN or carbapenem-VAN TZP–VAN vs. TZP	TZP–VAN vs. FEP or carbapenem-VAN: OR, 2.68; 95% CI, 1.83–3.91 TZP–VAN vs. VAN: OR, 3.40; 95% CI, 2.57–4.50	Time to AKI for TZP–VAN vs. FEP–VAN or carbapenem: mean difference, −1.30; 95% CI, −3.00 to 0.41 d).
Ciarambino T, et al., [44]	2020	6 (Only studies including adult patients)	9672	2 June 2019	TZP–VAN vs. VAN alone	OR, 2.77 (95% CI 1.94, 3.96); *p* < 0.0001	Not provided
Bellos I, et al., [12]	2020	47 (37 included only adults and 10 included only children)	56,984	20 August 2019	TZP–VAN vs. VAN alone TZP–VAN vs. FEP–VAN TZP–VAN vs. MER–VAN	TZP–VAN vs. VAN: OR, 2.05; 95% CI, 1.17–3.46 TZP–VAN vs. MER–VAN: OR, 1.84; 95% CI, 1.02–3.10 TZP–VAN vs. FEP–VAN: OR, 1.80; 95% CI, 1.13–2.77	TZP–VAN insignificantly increased risk of severe AKI and requirement of RRT. Time to AKI, duration of AKI, recovery from AKI, length of hospitalization and mortality were similar between the comparison groups.
Alshehri AM, et al., [45]	2022	12 (Only studies including adult patients)	14,511	November 2021	TZP–VAN vs. MER–VAN	TZP–VAN vs. MER–VAN: OR, 2.31; 95%CI, 1.69–3.15	The secondary outcomes, including hospital length of stay, RRT, or mortality were similar between the two groups

Abbreviations: n, number; AKI, acute kidney injury; ICU, intensive care unit; VAN, vancomycin; TZP, piperacillin–tazobactam; FEP, cefepime; MER, meropenem; OR, odds ratio; CI, confidence interval; RRT, renal replacement therapy.

## Data Availability

Not applicable.

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
