# Peer review of "Piperacillin–Tazobactam Plus Vancomycin-Associated Acute Kidney Injury in Adults: Can Teicoplanin or Other Antipseudomonal Beta-Lactams Be Remedies?"

_healthcare, 2022, doi:10.3390/healthcare10081582_

Round 1

Reviewer 1 Report

The manuscript ID healthcare-1824385 entitled "Piperacillin-tazobactam plus vancomycin-associated acute kidney injury in adults: can teicoplanin or other antipseudomonal beta-lactams be a remedy?" is an interesting review. Numerous observational studies and meta-analyses have suggested that combination therapy consisting of piperacillin-tazobactam (TZP) and vancomycin (VAN) augments acute kidney injury (AKI) risk when compared to viable alternatives such as cefepime-vancomycin (FEP-VAN) and meropenem-VAN. However, the exact pathophysiological mechanism(s) of this phenomenon is still unclear. One major limitation of the existing studies is the utilization of serum creatinine to quantify AKI since serum creatinine is not a sufficiently sensitive and specific biomarker to truly define the causal relationship between TZP-VAN exposure and nephrotoxicity. Even so, some preventive measures can be taken to reduce the risk of AKI when TZP-VAN is preferred. These measures include limiting the administration of TZP-VAN to 72 hours, choosing FEP-VAN in place of TZP-VAN in appropriate cases, monitoring the VAN area under curve level rather than VAN trough level, avoidance of exposure to other nephrotoxic agents, minimizing the prescription of TZP-VAN for patients with a high risk of AKI. More data are needed to make any comment on the beneficial impact of an extended-infusion regimen of TZP on nephrotoxicity. Additionally, TZP and teicoplanin can be a reasonable alternative to TZP-VAN for the purpose of declining AKI risk. However, the data are scarce to advocate this practice convincingly.

I appreciate the authors for this good review. However, the following changes are required to get a wider-scientific audience. 

In general, the review article should emphasize great ongoing limitations and possible alternatives to fix them. Here, the authors narrate more text without figures or mechanism diagrams, proper statistical analysis makes it a bit harder to evaluate the scientific soundness of the study. The figures and statistical analysis express the level of understanding by the authors on this topic and attract more guidance as well. 

I recommend the authors revise the current form accordingly. 

Author Response

We would like to thank all the reviewers for their valuable comments and contributions.

The aim of this narrative literature review is to summarize the published information on the risk of AKI caused by TZP-VAN combination regimen and other alternative regimens including FEP-VAN and MER-VAN. Furthermore, we reviewed the literature comparing the incidence of AKI in TZP-VAN and TZP-TEI regimens. As this study is a narrative review, performing statistical analysis is not compatible with the format of this work. With respect to your request, we have already depicted the results of all published systematic review and meta-analyses comparing the risk of AKI in TZP-VAN with those of either VAN alone or VAN plus anti-pseudomonal beta-lactam regimens. In these analyses, there is a significant heterogeneity among included studies and great majority of the studies had retrospective single-center design. Considering the lack of randomized controlled trial in this field, these meta-analyses may have a potential to augment bias. In addition, there are only 2 retrospective cohort studies comparing the frequency of AKI in TZP-VAN and TZP-TEI groups. Therefore, performing a statistical analysis including only 2 retrospective cohort studies would not be informative and we need more well-designed observational studies and randomized controlled trials in this area.

In response to your request, we added a figure showing the organic anion transporter-mediated mechanism of serum creatinine increase in TZP-VAN combination therapy.

Reviewer 2 Report

This is a timely and extensive review on the role of antibiotics in AKI. This is well written and researched. 

I would include a section to clarify how many studies looked at critically ill patients/ ICU patients. This data is presented in the table and would be better served with an expanded discussion.

Author Response

Thank you very much for your evaluation. In line with your suggestion, the risk of AKI in ICU patients receiving TZP-VAN and its comparison with other combinations (i.e., FEP-VAN and MER-VAN) were discussed in detail in a separate section.

Reviewer 3 Report

The authors cover all the previous studies, and the review is well systematized and comprehensively described. 

Author Response

Thank you very much for taking your precious time to evaluate this work.

Round 2

Reviewer 1 Report

The revised form of Manuscript ID healthcare-1824385, entitled 

"Piperacillin-tazobactam plus vancomycin-associated acute kidney injury in adults: can teicoplanin or other antipseudomonal beta-lactams be a remedy?" have the following issues. 

In general, if the author validates the pros and cons of any drug, they need to perform a meta-analysis using observational studies and randomized clinical trials. This is the most feasible way because the drugs are already in practice. 

I am curious, how does the narrative review validate or provide solid evidence to practice the mentioned drug in the proposed manner?

As the author mentioned, "performing a statistical analysis including only 2 retrospective cohort studies would not be informative and we need more well-designed observational studies and randomized controlled trials in this area". It seems, the current need more well-design with proper issues in the limitation of the meta-analysis.  

I am curious, how did the authors include the table 1 and 2? No proper description. 

 From the figure, I would like to know, whether creatinine is secreted in the tubular lumen or transferred from blood?

Where is the initial creatinine from blood or kidney or another organ? 

What is the link between the basolateral and the apical membrane?

Which one is pathologic condition and normal?

Author Response

We thank the reviewer for the careful evaluation and questions and comments. We believe that the following revisions by responding the reviewer’s comments will strengthen the quality of our manuscript.

C1- In general, if the author validates the pros and cons of any drug, they need to perform a meta-analysis using observational studies and randomized clinical trials. This is the most feasible way because the drugs are already in practice.

R1- There is no RCT comparing the AKI risk with TZP-VAN and anti-pseudomonal beta-lactams plus VAN. This holds true for TZP-TEI vs TZP-VAN combination regimens. In this narrative review, we aimed to assess the existing data. There are already 7 published systematic review and meta-analyses addressing TZP-VAN regimen-related AKI. As experts in this field, we tried to convey our point of view in this area by boiling down the literature.

C2- I am curious, how does the narrative review validate or provide solid evidence to practice the mentioned drug in the proposed manner?

R2- You always emphasize the potential advantages of systematic review and meta-analysis. But narrative reviews provide very important information. If this was not the case, no narrative reviews would be worth to be published by academic journals. Unlike meta-analyses, which have very narrowly defined parameters and precise inclusion and exclusion criteria, a narrative review ensures to touch upon various aspects of a topic. The narrative review provides more potential for individual insight and opportunities for speculation than most meta-analysis approaches.

C3- As the author mentioned, "performing a statistical analysis including only 2 retrospective cohort studies would not be informative and we need more well-designed observational studies and randomized controlled trials in this area". It seems, the current need more well-design with proper issues in the limitation of the meta-analysis.

R3- In this narrative review, we also evaluated the studies investigating TZP-TEI combination regimen-related AKI. As TEI has similar spectrum of activity with VAN, TZP-TEI can be a reasonable alternative to TZP-VAN if the rate of AKI is lower in the former combination. For this reason, we added this part to the narrative review and underlined the limitations of published studies and encourage investigators to perform well-design studies in this field. As an example, at the end of this review, we summarize one of our ongoing studies comparing the risk of AKI in patients receiving TZP-VAN with patients receiving TZP-TEI.’’

C4- I am curious, how did the authors include the table 1 and 2? No proper description. 

R4- We gave details in page 2 line 86-87 where you can see the description as follows ‘’ The results of the studies are summarized in Table 1 [9, 10, 18-41].’’

Similarly, the description can be found in page 3 line 115-116 for table 2 as follows ‘’Nevertheless, 7 meta-analyses have been reported to address the relationship between TZP-VAN and AKI as depicted in Table 2 [5,12,13,42-45].’’

C5- From the figure, I would like to know, whether creatinine is secreted in the tubular lumen or transferred from blood?

R5- It is obviously secreted into tubular lumen.

C6- Where is the initial creatinine from blood or kidney or another organ? 

R6- The initial creatinine is from blood. The creatinine is transported to kidney through blood flow and secreted into tubular lumen in this figure.

C7- What is the link between the basolateral and the apical membrane?

R7- The basolateral membrane is the tubular epithelium membrane facing to blood vessels. On the other side, apical membrane is facing to tubular lumen. After creatinine is transported into tubular epithelial cell, it is secreted into tubular lumen by specific transporters.

C8- Which one is pathologic condition and normal?

R8- There is no pathologic or normal condition comparison in this figure. This figure shows how TZP and VAN competitively inhibit secretion of creatinine into tubular lumen. The first mechanism is limiting the transport of creatinine into the tubular cell by binding of TZP to OAT1 and OAT3. The second is the inhibition of the expression of mRNA involved in OAT1-OAT3 externalization by VAN.